# A Window into the Workings of *anti*-B_18_H_22_ Luminescence—Blue-Fluorescent Isomeric Pair 3,3′-Cl_2_-B_18_H_20_ and 3,4′-Cl_2_-B_18_H_20_ (and Others) [note 1]

**DOI:** 10.3390/molecules28114505

**Published:** 2023-06-01

**Authors:** Marcel Ehn, Dmytro Bavol, Jonathan Bould, Vojtěch Strnad, Miroslava Litecká, Kamil Lang, Kaplan Kirakci, William Clegg, Paul G. Waddell, Michael G. S. Londesborough

**Affiliations:** 1Institute of Inorganic Chemistry of the Czech Academy of Sciences, 250 68 Řež, Czech Republic; ehn@email.cz (M.E.); bavol@iic.cas.cz (D.B.); jbould@gmail.com (J.B.); strnvoj@seznam.cz (V.S.); litecka@iic.cas.cz (M.L.); lang@iic.cas.cz (K.L.); kaplan@iic.cas.cz (K.K.); 2Department of Physical Chemistry, University of Chemistry and Technology in Prague, Technická 5, Dejvice, 166 28 Prague, Czech Republic; 3Chemistry, School of Natural and Environmental Sciences, Newcastle University, Newcastle upon Tyne NE1 7RU, UK; bill.clegg@newcastle.ac.uk (W.C.); paul.waddell@newcastle.ac.uk (P.G.W.)

**Keywords:** *anti*-B_18_H_22_, luminescence, fluorescence, cluster boron hydrides, chlorination, halogenation, substitution, excited-state lifetime, quantum yield

## Abstract

The action of AlCl_3_ on room-temperature tetrachloromethane solutions of *anti*-B_18_H_22_ (**1**) results in a mixture of fluorescent isomers, 3,3′-Cl_2_-B_18_H_20_ (**2**) and 3,4′-Cl_2_-B_18_H_20_ (**3**), together isolated in a 76% yield. Compounds **2** and **3** are capable of the stable emission of blue light under UV-excitation. In addition, small amounts of other dichlorinated isomers, 4,4′-Cl_2_-B_18_H_20_ (**4**), 3,1′-Cl_2_-B_18_H_20_ (**5**), and 7,3′-Cl_2_-B_18_H_20_ (**6**) were isolated, along with blue-fluorescent monochlorinated derivatives, 3-Cl-B_18_H_21_ (**7**) and 4-Cl-B_18_H_21_ (**8**), and trichlorinated species 3,4,3′-Cl_3_-B_18_H_19_ (**9**) and 3,4,4′-Cl_3_-B_18_H_19_ (**10**). The molecular structures of these new chlorinated derivatives of octadecaborane are delineated, and the photophysics of some of these species are discussed in the context of the influence that chlorination bears on the luminescence of *anti*-B_18_H_22_. In particular, this study produces important information on the effect that the cluster position of these substitutions has on luminescence quantum yields and excited-state lifetimes.

## 1. Introduction

With the obvious exception of carbon, it is boron that amongst all the elements boasts the greatest number and diversity of hydride compounds [1]. These species produce polyhedral cluster assemblies with an ostensible 12-vertex icosahedral size limit. This size constraint, however, can be surpassed by the intimate fusion of two or more clusters to form ‘macropolyhedral’ species, in which constituent subclusters conjoin via shared polyhedral edges or faces [2]. We are interested in how the chemical modification of such macropolyhedral boron hydride clusters produces changes in their physical properties, and in particular, with regard to the interaction of these molecules with laser light. At very high excitation energies, our investigations hold relevance to the generation of boron–proton plasmas, and, at lower excitation energies, emission from electronically excited states that form the basis for an important class of light sources. Here, there is a renewed interest in the binary boron hydride cluster compound *anti*-B_18_H_22_ **1** due to the discovery of its utility as a new class of laser material [3]. It is also of particular interest as it fluoresces in the elusive blue region of the spectrum, with a quantum yield approaching unity [4]; it is highly photostable [3], and it is readily soluble in organic polymer matrices [3,5]. These are all factors of general practical benefit regarding the fabrication of optical devices. The photophysics of **1** may be modified by replacing some of its cluster terminal hydrogen atoms with substituents or ligands such as –SH [6], bromine [7,8], iodine [9], alkyls [10], and pyridine [11,12] to give molecules capable of, for example, environment-sensitive thermochromic luminescence [11], single-molecule multiple emissions [7], and the photosensitisation of oxygen [9]. Recently, metal-catalysed and metal-free nucleophilic substitution of the iodine group in 7-I-B_18_H_21_ [9,13] has been demonstrated [14], leading to a series of luminescent B-N, B-O, and B-S substituted octadecaborane derivatives [14]. In addition, a monochlorinated derivative of **1**, the blue fluorescent 7-Cl-B_18_H_21_, has been reported, featuring a quantum yield of luminescence of 0.8 [15]. Collectively, these entries to the literature are the beginnings of a systematic survey of the functionalisation of **1**, from which there are starting to emerge several broader patterns linking the photophysical behaviour of these molecules to their chemical structure. Accompanying these trends are tentative explanations based on, for example, the principles of ‘Total Cluster Volume’ [10], cluster electron density mapping [10], and the cluster-position of substitution [14]. Here, we report an extension to this growing portfolio of compounds by the synthesis of several chlorinated derivatives of **1**, the highest yielding examples of which are blue-fluorescent isomers 3,3′-Cl_2_-B_18_H_20_ (**2**) and 3,4′-Cl_2_-B_18_H_20_ (**3**). Compounds **2** and **3** are the first examples of halogenation of **1** at the B3 or B3′ positions, and thus afford further opportunity to deepen the understanding on what are the chemical factors influencing luminescence from *anti*-B_18_H_22_ and its derivatives.

## 2. Results and Discussion

**Synthesis and structural characterization.** Addition of aluminium chloride to room-temperature tetrachloromethane solutions of *anti*-B_18_H_22_ **1** results primarily in the formation of a mixture of two of its dichlorinated derivatives, isomers 3,3′-Cl_2_-B_18_H_20_ (**2**) and 3,4′-Cl_2_-B_18_H_20_ (**3**) (Figure 1), which, after work-up and purification via sublimation, were obtained in an non-optimised yield of 76%. The mass spectrum of the crude product mixture (Appendix A) indicates the additional presence of lesser amounts of mono- and trichlorinated derivatives of **1**. The generally smaller quantities of these species precluded their full examination, but we were able to collect good NMR, UV-vis, and crystallographic data on some of these minor products that we present later in this manuscript. Temperature and duration of reaction seem to bear an influence on both the rate at which the chlorinated derivatives are formed as well as the ratio in which they are produced. Thus, typically, a room temperature reaction requires a period of 4–5 days for the complete consumption of **1**. The same synthesis carried out at 55 °C gives a full conversion to products in only 90 min, whereas at an ice-bath temperature of 2 °C a period of 30 days is required. The product mix ratios (Table 1) were quantitatively determined by analytical HPLC and an analysis of the ^11^B NMR spectra of the reaction mixtures and, in particular, a comparison of the integrations of the singlets at δ(^11^B) +23.1 ppm for the substituted B3 and/or B3′ position(s), and at −24.3 ppm for the substituted B4 and/or B4′ position(s) (Appendix A). These ratios show that at higher temperatures, substitution at the B4 and/or B4′ position(s) becomes more probable, and the formation of compound **3** and trichlorinated derivatives **9** and **10** are therefore more likely. Lower temperatures or shorter reaction times boost the amount of monochlorinated derivatives **7** and **8** in the product mix (primarily **7**), and longer reaction times, over a certain temperature, support trichlorination and the formation of **9** and **10** (primarily **9**). 

HPLC, fractional sublimation, and crystallization were used to separate the various products of chlorination of **1**. Our discussion of these new species begins with the two main products, dichlorinated **2** and **3**.

The molecular structures of **2** and **3**, as determined by single-crystal X-ray diffraction studies (SCXRD), are shown in Figure 1. Both isomers preserve the classical octadecaborane polyhedral architecture of **1**, with no perturbation of significance in any of the B-B connectivities. Equally, there is very little difference between B-Cl bond lengths for the different substituent positions that all measure approximately 1.8 Å, which is a similar value to B-Cl bonds in other known macropolyhedral borane systems [16,17]. Isomers **2** and **3** provide identical mass spectra with *m*/*z* 284.32, due to their [B_18_H_19_Cl_2_]^−^ ions (Appendix A), but may be distinguished by their different absorption spectra and retention times when passed through an HPLC system prior to injection into the mass spectrometric device (Appendix A). The relative positions of substitution in isomers **2** and **3** result in the preservation of *C_i_* symmetry in **2** and asymmetry in **3**. These differences are clearly seen in the NMR data for the compounds (see Appendix A vs. Appendix A for direct comparison, and Figure 2 for wider comparison with all chlorinated derivatives of **1** described in this manuscript).

Table 2 shows a listing of the measured and calculated NMR data for compound **2** that are entirely consistent with its *C_i_* symmetric molecular structure shown in Figure 1. The excellent correlation between the measured data and those calculated for the molecular configuration shown in Figure 1 provide further weight to this structure being representative of the bulk sample of **2**. The degree of perturbation from the δ(^11^B) chemical shifts for parent compound **1** is small for **2**, with only the chlorine-substituted B(3,3′) chemical shift changing significantly (Figure 3). The chlorine atoms deshield the B(3,3′) nuclei, shifting the resonance downfield by about 10 ppm.

Table 3 lists the measured and calculated NMR data for compound **3**. The asymmetry of **3** is immediately apparent in its ^11^B NMR spectrum, which displays 18 peaks. These peaks may be conveniently split into two sets of nine resonances for each of the molecule’s two subclusters: {3-Cl-B_9_} and {4′-Cl-B_9_}. The nine peaks for the {3-Cl-B_9_}-subcluster essentially overlap with those for the symmetrical compound **2**. The {4′-Cl-B_9_}-subcluster, on the contrary, displays change; the B3′-H peak shifts upfield by about 9 ppm to a position similar to that in parent compound **1**, and there is a roughly 16 ppm downfield shift of the B4′-Cl peak (see dotted lines in Figure 3).

Overall, the ^11^B NMR properties of compounds **2** and **3** suggest them to be the same two dichlorinated *anti*-B_18_H_20_Cl_2_ species mentioned, but structurally undefined and uncharacterised, in a previous report describing the chlorination of **1** with SO_2_Cl_2_ [15].

The charge distribution in *anti*-B_18_H_22_ is such that the basal boron atoms B(1), B(2), B(3) and, in particular, B(4) (see Figure 4, and Figure 1 for numbering) possess the greatest share of the available electron density. This distinction is apparent in the products of electrophilic substitution of **1** recorded in the literature so far. Thus, it is the case for recently published brominated and iodinated derivatives of **1** that, under similar reaction conditions to those used here, bromination results mainly in the monohalogenated product 4-Br-B_18_H_21_ (although a partially characterised dibrominated derivate 4,4′-Br_2_-B_18_H_20_ was also observed in mass spectrometric analysis and NMR) [7], and iodination to high yields of the dihalogenated species 4,4′-I_2_-B_18_H_20_. (n.b. the reaction of the dianionic form of **1** with iodine in alcohol solutions gives the monosubstituted 7-I-B_18_H_21_) [9]. It is therefore apparent from this work that the chlorination of **1** is far more versatile, and the substitution of its octadecaborane cluster is more multi-directed than is the case for bromination or iodination, in particular with regard to the activation of the B3 cluster position towards substitution. The relatively lower energy of the 3,3′-Cl_2_-B_18_H_20_ isomer **2**, calculated at 2.3 kcal/mol, would, all other things being equal, lead to a Boltzmann distribution of 98% for **2** and 2% for isomer **3**. However, the considerably more negative charge associated with the B(4 and 4′) cluster atoms (see Figure 4) presumably competes with the differences in zero-point total energies, especially at higher temperatures, leading to the observed ratios of **2**:**3** formed during synthesis at 55 °C.

Above and beyond the two main products from this reaction, compounds **2** and **3**, an effort was made to isolate and characterise the lower-yielding side products. Laborious work involving HPLC separation and identification using mass spectrometry and UV absorption spectroscopy resulted in the imperfect isolation of an additional six chlorinated derivatives of **1**. Of these six compounds, three are further isomers of dichlorinated derivatives of **1**, specifically 4,4′-Cl_2_-B_18_H_20_ (**4**), 3,1′-Cl_2_-B_18_H_20_ (**5**), and 7,3′-Cl_2_-B_18_H_20_ (**6**). The molecular structures of each of these isomers were established by SCXRD analyses, the results of which are shown in Figure 5.

These molecular structures were supported by mass spectrometry and, in the cases of **4** and **5**, a comparison of measured NMR data with calculated NMR chemical shifts for specific optimised molecular structures (Table 4 and Table 5). Unfortunately, only ^11^B NMR data were collected for compound **6**, which was isolated as a single crystal in one of many fractional sublimation procedures used to purify other dichlorinated species. These measured data nevertheless match well with those calculated for the ^11^B chemical shifts (Table 6) of a molecular geometry for **6** shown in Figure 5.

Two mono-chlorinated derivatives of **1**, 3-Cl-B_18_H_21_ (**7**) and 4-Cl-B_18_H_21_ (**8**), were also observed. Although the isolation of these monochlorinated derivatives from other, multichlorinated, compounds was facile, the complete separation of **7** from **8** proved too difficult. However, SCXRD studies on the solid solutions of these compounds resulted in the molecular structures shown in Figure 6. Again, the octadecaborane cluster structure of **1** is unperturbed; however, the B-Cl distances appear to be about 0.1 Å shorter than those lengths for the dichlorinated species described here, although this may be primarily due to the disorder present in the solid solution.

Analytical chromatography of the multi-chlorinated mixture led to the separation of compounds **7** and **8** from the other multi-chlorinated species. Initial separations generally gave a mixture of **7** and **8** with a mass spectrum of *m/z* 250.32, due to their [B_18_H_20_Cl]^−^ ions (Appendix A). NMR of these mixtures indicated the presence of about 85% of 3-Cl-B_18_H_21_ (**7**) to 15% of 4-Cl-B_18_H_21_ (**8**), based on an integration of the B3-Cl singlet peak (located at δ(^11^B) +23.2 ppm) for the former and the B4-Cl singlet peak (located at −24.5 ppm) for the latter compound. Crystals (of the three samples measured by SCXRD) of this sample that were obtained via sublimation show, however, a reversal of this ratio (approximately 75% of **8** to 25% of **7**, with small variations in different crystals), suggesting that **8** is the more volatile of the two isomers. Indeed, this physical difference enabled the effective separation of **8** from **7** via careful sublimation and the isolation of the 3-Cl derivative in good purity for NMR and UV-vis studies. There is a very good correlation between measured NMR data for **7** and chemical shifts calculated for the monochlorinated 3-Cl-B_18_H_21_ structure (Table 7). 

Finally, an attempt was made to determine the location of the chlorine substituents on the very small amount of isolated trichlorinated species using a comparison of the collected experimental ^11^B NMR spectroscopic data with the computed chemical shifts for a series of seven trichlorinated isomers with the lowest zero-point corrected total energies, as shown in Table 8.

Appendix A shows the measured ^11^B NMR spectrum of the HPLC-separated fraction with a single mass spectrometric peak at *m*/*z* 319.22 (Appendix A), which computes precisely to the expected mass of [Cl_3_-B_18_H_18_]^−^. This spectrum appears to comprise two molecular species present in an approximate 2:1 ratio. Appendix A and Appendix A are the calculated spectra for all of the seven lowest energy trichlorinated isomers listed in Table 8. A comparison of these measured and calculated data suggest that the main component is very likely 3,4,3′-Cl_3_-B_18_H_19_ (**9**) and the minor component most probably 3,4,4′-Cl_3_-B_18_H_19_ (**10**). However, in absence of any reliable crystallographic data, these are tentative assignments. The measured and calculated NMR data for 3,4,3′-Cl_3_-B_18_H_19_, compound **9**, are shown in Table 9.

A detailed comparison of the molecular geometry of these various chlorinated clusters, together with the parent *anti*-B_18_H_22_ (**1**) [18], shows that the B–B (and B–Cl) bond lengths are rather insensitive to the substituent positions. Because structural disorder has an averaging effect on parts of the molecules in which disorder is not resolved and twinning adversely affects precision, such an analysis can be carried out reliably only on ordered structures, which here means one form of compound **2** (not the forms in the Supporting Information), and compounds **3**, **4**, and **6**; two of these (**2** and **4**) have more than one crystallographically independent molecule, all of which can be included in the comparison. A full list of bond lengths and their differences from the corresponding bonds in compound **1** are given in Appendix A. These differences between the substituted and parent clusters range from −0.045 Å to +0.034 Å, while the total range of all B–B bond lengths is 0.294 Å (from 1.706 to 2.002 Å, with individual crystallographic standard uncertainties between about 0.002 and 0.02 Å); they follow no clear pattern relative to the positions of substitution. We therefore conclude that the cluster bonding is essentially unaffected by chlorination.

**Photophysical properties of some of the chlorinated derivatives of 1**. As mentioned in the Introduction, compound **1** [3,4,5] and its derivatives [6,7,8,9,10,11,12,14,15,16] display various modes of useful luminescence. As such, we were keen to understand the photophysics of the new chlorinated derivatives of **1** reported here. Table 10 and Figure 7 show the main photophysical data recorded for new compounds **2**, **4**, and **7**. Absorption and excitation spectra for these species may be found in the SI (Appendix A). Overall, there is little difference in the absorption and emission maxima for these derivatives, which all absorb in the UV-A region (absorption maxima in the 324–344 nm range) and emit in the blue region (emission maxima in the 406–435 nm range). Thus, chlorination seems to have only a small influence over the relative energies of the contributing HOMO and LUMO systems. However, from these data a trend is apparent that increasing the number of chlorine substituents both shortens the lifetimes of the fluorescent excited states (*τ_L_*) and decreases the quantum yield of fluorescence (*Φ_L_*). So, for example, mono-chlorinated derivative **7** has about half of the fluorescent lifetime of parent compound **1** and approximately half of its *Φ_L_*, and dichlorinated **2** has just over a tenth of the fluorescent lifetime of **1** and roughly a twelfth of its *Φ_L_*. 

A recent article by Spokoyny et al. [15] proposes that the location of substituents on the octadecaborane cluster plays a crucial role in the efficiency of emission from the molecule (its quantum yield of luminescence). In their paper [15], a comparison was made between 7-I-B_18_H_21_ and 4-I-B_18_H_21_ isomers, with the former displaying a two-fold greater quantum yield of phosphorescence. The mono-chlorinated species 7-Cl-B_18_H_21_ (**11**) is also reported, the photophysical properties of which are also shown here in Table 10. Comparison of the data for **7** and **10** reveals that chlorination of the basal B3 position in **7** leads to a roughly 25% shorter fluorescent lifetime and 25% lower quantum yield than in **11**, where chlorination is at the open-face ‘gunwale’ B7 position. A further interesting comparison is between the photophysics of compounds **2** and **4**: although both species have a similarly short excited-state lifetime, compound **4** has more than twice the quantum yield of fluorescence than **2**. This suggests that chlorination (and by tentative extrapolation, substitution) of **1** at the B4/4′ positions reduces the efficiency of fluorescence less than when at the B3/3′ cluster sites.

## 3. Experimental

### 3.1. Materials

*Anti*-B_18_H_22_ was made using the oxidative fusion of the nonaborane anion made from *nido*-B_10_H_14_, as described by Gaines et al. [19]. *nido*-B_10_H_14_ purchased from Katchem s.r.o. (Kralupy, Czech Republic) and AlCl_3_ (Sigma Aldrich, Merck Life Science spol. S r.o. Prague, Czech Republic) were both sublimed before use; CCl_4_, *n*-hexane, and CH_2_Cl_2_ (all Lach-ner, Prague, Czech Republic) were used without purification. Silica gel for chromatography was 60–200 μm (Lach-ner, Prague, Czech Republic). The authors recommend that, prior to any use of *nido*-B_10_H_14_, a consultation of the information on its toxicity and required safety precautions are made (see https://cameochemicals.noaa.gov/chemical/503#es and https://inchem.org/documents/icsc/icsc/eics0712.htm, accessed on 2 May 2023).

### 3.2. Chlorinated Derivatives of Anti-B_18_H_22_ (1)

A 100 mL RB-flask fitted with a Teflon stop-cock was charged with *anti*-B_18_H_22_ (360 mg, 1.7 mmol) and freshly sublimed AlCl_3_ (2.7 g, 20 mmol), to which was added 20 mL of CCl_4_ under a stream of argon gas. The vessel was then closed, and the contents stirred. The reaction was monitored by NMR spectroscopy at regular intervals. On the complete consumption of the *anti*-B_18_H_22_ starting material (4–5 days at RTP or approx. 90 min at 55 °C), the reaction vessel was placed in an ice-bath, allowed to cool, and then cold distilled water (25 mL) that had been acidified with a small volume of HCl_(aq)_ (3 mL) was added slowly whilst stirring so as to destroy any remaining AlCl_3_. The two-layer liquid was then filtered, and the water layer separated and extracted three times with 15 mL measures of dichloromethane. The combined CCl_4_ and CH_2_Cl_2_ solutions were then reduced to dryness on a rotary evaporator and the remaining solids sublimed at 120 °C. Subsequent crystallisation from a minimum amount of hot hexanes gave 3,3′-Cl_2_-B_18_H_20_ (**2**) and 3,4′-Cl_2_-B_18_H_20_ (**3**) jointly in a 76% yield (373 mg, 1.31 mmol). The remaining minor components were isolated, as described in the manuscript, using HPLC techniques.

### 3.3. General Methods

Preparative column chromatography was carried out using a 4 cm diameter column of silica gel G (60–200 μm Lach-ner). **NMR** spectra were recorded on a JEOL 600 MHz (14.1 T) spectrometer. Structural assignments were made using ^11^B, ^11^B-{^1^H}, ^1^H, ^1^H-{^11^B(broadband)}, ^1^H-{^11^B(selective)}, COSY, and HMQC (Heteronuclear Multiple-Quantum Correlation) techniques. **Mass Spectrometry** measurements were performed on a Thermo-Finnigan LCQ-Fleet Ion Trap instrument using electrospray ionization (ESI) with the detection of negative ions. For ESI, samples dissolved in acetonitrile (concentrations approximately 100 ng·mL^−1^) were introduced to the ion source by infusion at 5 μL·min^−1^ with a source voltage of 4.5 kV, a tube lens voltage of −90.8 V, a capillary voltage of −20.3 V, a capillary temperature of 275 °C, and a nebulizing sheath gas and auxiliary gas flow of 1.8 L·min^−1^ and 5.9 L·min^−1^, respectively. Molecular ions [*M*]^−^ were detected for all univalent anions as base peaks in the spectra. The experimental and calculated isotopic distribution patterns were fully agreed upon for all isolated compounds. The isotopic distribution in the boron plot of all peaks was in complete agreement with the calculated spectral pattern. The data are presented for the most abundant mass in the boron distribution plot (100%) and for the peak on the right side of the boron plot corresponding to the *m/z* value. **Analytical chromatography** was carried out on a Thermo Finnigan Surveyor HPLC system equipped with a Photo Diode Array detector. Chromatographic conditions: column LiChroCART RP-select B 60 Å (5 µm, 250 × 3.00 mm *I.D.*). Mobile phase preparation: 9 mL of 0.5 M stock solution of propylamine acetate (PAA) was diluted by water (HPLC grade) to 430 mL (pH was adjusted to 4.5). This buffer was mixed with 570 mL of ACN (HPLC gradient grade). Flow rate 0.4 mL∙min^−1^, detection PDA (190–800 nm). Samples with a concentration of approximately 1 mg∙mL^−1^ in the mobile phase were injected in a volume of 3 µL. ***Semi*-preparative chromatography** was realized on LaPrep Sigma, VWR system equipped with multiple-wavelength detector and a fully automated fraction collector Foxy R2, Teledyne ISCO. Chromatographic conditions: column TESSEK Separon SGX C18 (7 µm, 250 × 25.00 mm *I.D.*). Mobile phase preparation: 12 mL of 1.0 M stock solution of propylamine acetate (PAA) was diluted by water (HPLC grade) to 300 mL (pH was adjusted to 4.2). This buffer was mixed with 700 mL of ACN (HPLC gradient grade). Flow rate 8.0 mL∙min^−1^ and fixed wavelengths: 225, 335, and 355 nm. Samples with a concentration of approximately 10 mg∙mL^−1^ in the mobile phase were injected in a volume of 2 mL. **UV-vis absorption and fluorescence** properties were recorded on a Perkin Elmer Lambda 35 spectrometer and FLS1000 (Edinburgh Instrument, Livingston, UK) on the air-saturated solutions of the respective compounds. Fluorescence lifetime experiments were performed upon excitation at 340 nm (EPLED-340, 340 ± 10 nm) and decay curves were fitted to exponential functions by the iterative reconvolution procedure of the Fluoracle software (v. 2.13.2, Edinburgh Instrument, Livingston, UK). Fluorescence quantum yields were measured using a Quantaurus QY C11347-1 spectrometer (Hamamatsu, Shizuoka, Japan).

### 3.4. X-ray Diffraction Analyses

Single crystals of all compounds were grown either by slow evaporation of concentrated solutions of the compounds in *n*-hexane in a glass vial or from the sublimation of small amounts sealed in an evacuated glass NMR tube and placed with the bottom third of the tube in a sand-bath maintained at ca. 90 °C. Single-crystal diffraction data were measured on Rigaku Oxford Diffraction XtaLAB Synergy diffractometers in Prague and Newcastle. Structures were solved with SHELXT [20] and refined with SHELXL [21], with the aid of the OLEX2 interface [22]. Table 11 gives the main crystal and refinement data for compounds **2**–**8**; corresponding information for polymorphic forms is given in Appendix A. In the various structures the asymmetric unit contains one-half, one, or more than one molecule, with some molecules lying on inversion centres. In some cases, this means an unsymmetrical molecule is disordered; lower-symmetry models were explored and rejected. For the co-crystallised compounds **7** and **8**, disorder of the 3 and 4 positions occurs together with disorder of the molecule over an inversion centre; three separate crystals from two sublimation experiments were examined and gave consistent results with only minor variations of the composition (only one result is given here). Crystals of **4** and **2a** (one of the polymorphs of **2**) were twinned, adversely affecting the precision of the structure of **4** (for which the resolved two twin components are probably an approximation to multiple twinning). Some structures were found to contain minor amounts of other isomers, involving further disorder; two of the four forms of the 3,3′ compound **2** (**2b** and **2c**) contain small amounts of the 3,4′ isomer, while there is some minor additional 4 and/or 4′ Cl substitution identified in the structure of the 3,1′ isomer **5**. In most cases, particularly for fully ordered structures, H atoms could be located in difference maps and refined freely; for less well-determined H atoms, restraints were applied to maintain approximately the same geometry as in the ordered structures.

## 4. Conclusions

As compared to the halogenation of **1** with bromine and iodine, its chlorination using AlCl_3_ in CCl_4_ produces a larger number of substituted derivatives with halogenation occurring in a greater variety of positions on the 18-vertex boron cluster. The specific sites of chlorination in the two main products, dichlorinated isomers **2** and **3**, seem to be driven by the competing factors of charge distribution in **1** and the relative zero-point corrected total energies of the products themselves. Regarding the photophysics of these compounds, chlorination of **1** appears to reduce the excited-state lifetime of fluoresence as well as its quantum yield of fluorescence. These reductions are not linear, with dichlorination resulting in disproportionately greater reduction compared to monochlorination. Although the current sample size is still small, the data also suggest that substitution at the B4 position leads to a greater bathochromic shift in the emission wavelength compared to B3-substituted species, but it dampens the fluorescence less. Furthermore, although substitution at the ‘gunwale’ open-face B7 position is less stable in comparison to substitution at the basal B3 or B4 positions, it does lead to superior quantum yields of fluorescence.

In closing, it is evident that, whereas the profound reduction in the lifetime of excited states of these molecules caused by chlorination is detrimental to the system’s fluorescence quantum yield, it would presumably reduce the probability of excited-state absorption (ESA) of the pump and the stimulated emission energy that has been shown to be the major mitigating factor to the better laser performance of compound **1** [23]. It is therefore conceivable that the combination of chlorine substituents (to reduce the excited-state lifetime) with alkyl substituents (to boost quantum yield) [10] could produce a B_18_-derivative molecular system with good emission properties and fewer ESA complications, and hence an interesting laser performance.

## Figures and Tables

**Figure 1 molecules-28-04505-f001:**
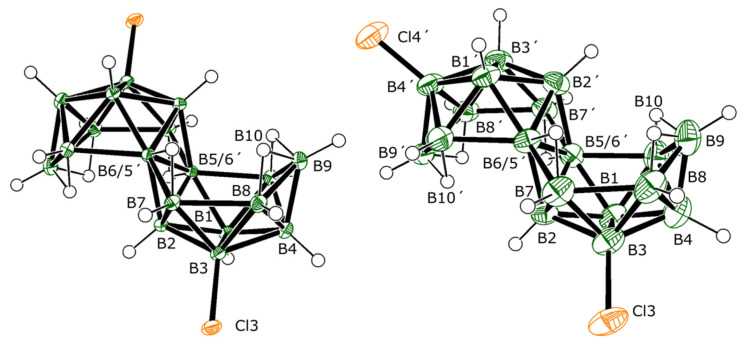
The crystallographically determined molecular structures of 3,3′-Cl_2_-B_18_H_20_ (**2**) and 3,4′-Cl_2_-B_18_H_20_ (**3**), drawn with 50% probability ellipsoids for non-H atoms. For a list of interatomic distances see Appendix A.

**Figure 2 molecules-28-04505-f002:**
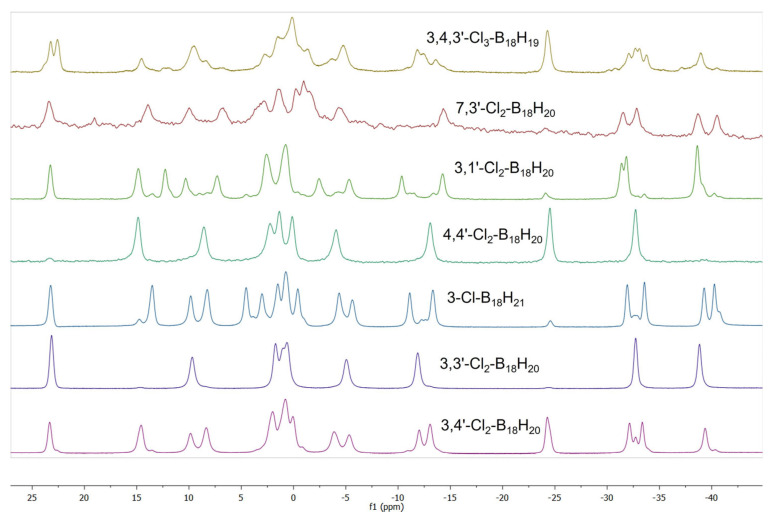
Stacked ^11^B-{^1^H} NMR spectra of all chlorinated derivatives of *anti*-B_18_H_22_ (**1**).

**Figure 3 molecules-28-04505-f003:**
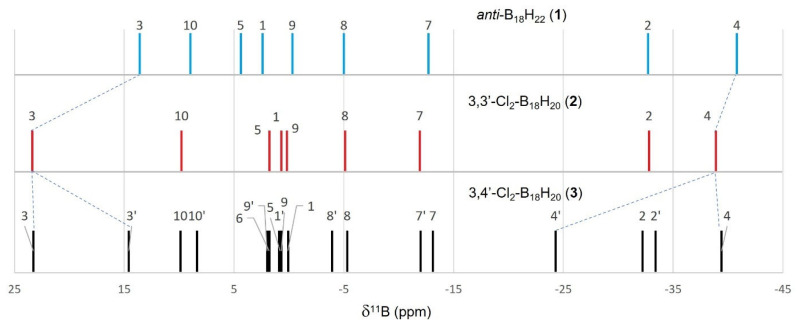
A comparison of ^11^B-{^1^H} NMR chemical shifts for *anti*-B_18_H_22_ (**1**), 3,3′-Cl_2_-B_18_H_20_ (**2**) and 3,4′-Cl_2_-B_18_H_20_ (**3**). Numbering for symmetrical **1** and **2** are simplified to single numbers, i.e., 3 = (3,3′) or 10 = (10,10′), etc.

**Figure 4 molecules-28-04505-f004:**
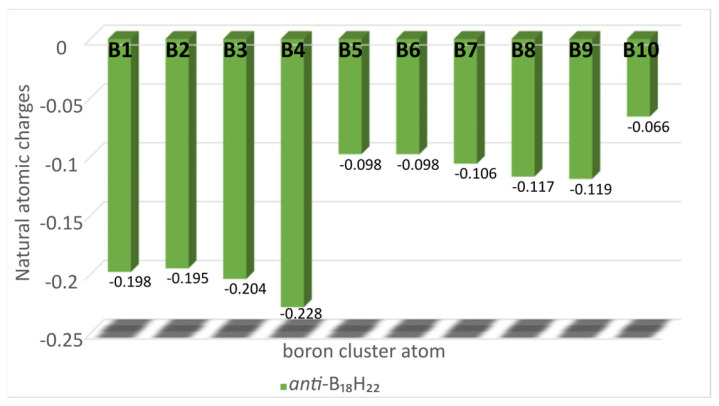
Natural atomic charges for boron positions in *anti*-B_18_H_22_ (**1**). Numbering for symmetrically equivalent atoms are simplified to single numbers, i.e., 3 = (3,3′) or 10 = (10,10′), etc.

**Figure 5 molecules-28-04505-f005:**
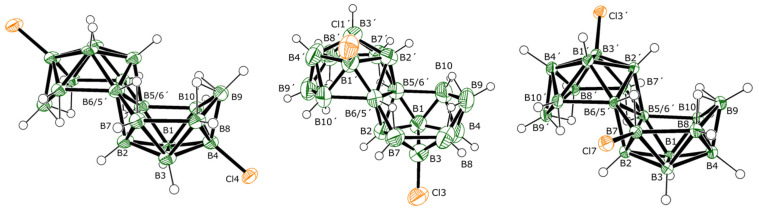
The crystallographically determined molecular structures of 4,4′-Cl_2_-B_18_H_20_ (**4**), 3,1′-Cl_2_-B_18_H_20_ (**5**), and 7,3′-Cl_2_-B_18_H_20_ (**6**), drawn with 50% probability ellipsoids for non-H atoms. For a list of interatomic distances see Appendix A.

**Figure 6 molecules-28-04505-f006:**
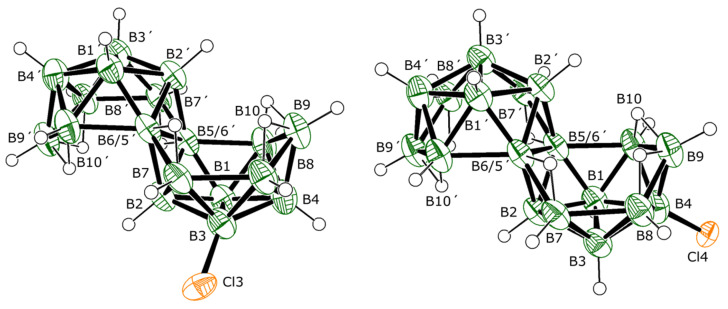
The crystallographically determined molecular structures of 3-Cl-B_18_H_21_ (**7**), and 4-Cl-B_18_H_21_ (**8**), drawn with 50% probability ellipsoids for non-H atoms. For a list of interatomic distances see Appendix A. The apparent angular distortion of the 3-Cl substituent is an artefact of disorder, which could be resolved only for the Cl atoms, so the experimentally determined boron atom positions are an average weighted towards the 4-Cl isomer.

**Figure 7 molecules-28-04505-f007:**
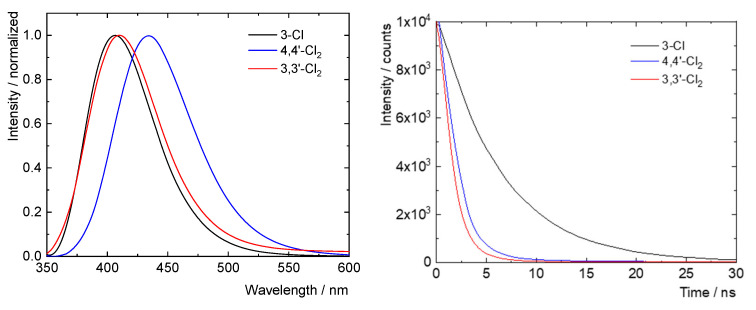
Normalised fluorescence spectra (**left**) and fluorescence decay spectra (**right**) for compounds **2**, **4** and **7**.

**Table 1 molecules-28-04505-t001:** The ratio of chlorinated derivatives of **1** formed using different reaction conditions. Percentages were calculated by the integration of peaks separated on HPLC (Apex retention times: 3.51 min for **9** and **10**; 7.18 min for **2**; 8.68 min for **3**; and 15.46 min for **7** and **8**). For more details on HPLC conditions, see Section 3.

React. Conditions	Monochlorinated Deriv. (7, 8)	3,3′-Cl_2_-B_18_H_20_ (2)	3,4′-Cl_2_-B_18_H_20_ (3)	Other Dichlorinated Deriv. (4 to 6)	Trichlorinated Deriv. (9, 10)
+2 °C/30 days	43%	20%	34%	<1%	3%
RTP/5 days	2%	41%	35%	<1%	21%
+55 °C/90 min	13%	29%	50%	<1%	8%

**Table 2 molecules-28-04505-t002:** Measured proton and boron-11 NMR data for 3,3′-Cl_2_-B_18_H_20_ (**2**) at 291 K in CDCl_3_ solution together with calculated values for comparison. (^11^B and ^11^B-{^1^H} NMR spectra are shown in Appendix A).

Assign.	δ(^11^B)/ppm		δ(^1^H)/ppm
	Meas.	Calc.	Meas.
3,3′	+23.4	+26.1	--- ^a^
10,10′	+9.8	+10.2	+4.06
5,6	+1.8	+1.9	--- ^b^
1,1′	+0.7	−0.1	+3.28
9,9′	+0.2	−0.2	+3.45
8,8′	−5.1	−4.8	+3.28
7,7′	−11.9	−11.2	+3.20
2,2′	−32.8	−33.0	−0.04
4,4′	−38.9	−38.3	+0.59
9,10		−1.28	−0.53
6,7		−1.28	−0.53
8,9		−3.05	−2.41

^a^ Chlorine substituent, ^b^ Site of conjunction.

**Table 3 molecules-28-04505-t003:** Measured proton and boron-11 NMR data for 3,4′-Cl_2_-B_18_H_20_ (**3**) at 291 K in CDCl_3_ solution together with calculated values for comparison. (^11^B and ^11^B-{^1^H} NMR spectra are shown in Appendix A).

Assign.	δ(^11^B)/ppm		δ(^1^H)/ppm
	Meas.	Calc.	Meas.
3	+23.3	+25.8	--- ^a^
3′	+14.6	+14.3	+4.12
10	+9.9	+9.0	+4.08
10′	+8.4	+7.5	+4.41
9′	+2.0	+1.0	+3.41
6	+1.8	−0.2	--- ^b^
1′	+0.9	−1.0	+3.41
5	+0.8	−1.2	--- ^b^
9	+0.7	−1.3	+3.31
1	+0.1	−2.6	+3.41
8′	−3.9	−6.1	+3.43
8	−5.3	−7.7	+3.30
7′	−12.0	−14.2	+3.04
7	−13.1	−15.3	+3.20
4′	−24.3	−24.1	--- ^a^
2	−32.2	−34.8	−0.01
2′	−33.4	−36.2	+0.02
4	−39.4	−42.2	+0.64
9′,10′			−0.07
9,10			−0.47
5,7′			−0.50
6,7			−1.07
8′,9′			−2.26
8,9			−2.44

^a^ Chlorine substituent, ^b^ Site of conjunction.

**Table 4 molecules-28-04505-t004:** Measured proton and boron-11 NMR data for 4,4′-Cl_2_-B_18_H_20_ (**4**) at 291 K in CDCl_3_ solution together with calculated values for comparison. (^11^B and ^11^B-{^1^H} NMR spectra are shown in Appendix A).

Assign.	δ(^11^B)/ppm		δ(^1^H)/ppm
	Meas.	Calc.	Meas.
3,3′	+14.6	+15.1	+4.13
10,10′	+8.4	+7.5	+4.42
5,6	+2.1	+0.2	--- ^a^
9,9′	+1.3	−1.0	+3.78
1,1′	+0.0	−1.1	+3.30
8,8′	−4.3	−6.5	+3.44
7,7′	−13.2	−14.9	+3.03
4,4′	−24.6	−23.6	--- ^b^
2,2′	−35.9	−35.9	+3.03
9,10			+0.03
6,7			−0.96
8,9			−2.18

^a^ Site of conjunction, ^b^ Chlorine substituent.

**Table 5 molecules-28-04505-t005:** Measured proton and boron-11 NMR data for 3,1′-Cl_2_-B_18_H_20_ (**5**) at 291 K in CDCl_3_ solution together with calculated values for comparison. (^11^B and ^11^B-{^1^H} NMR spectra are shown in Appendix A).

Assign.	δ(^11^B)/ppm		δ(^1^H)/ppm
	Meas.	Calc.	Meas.
3	+23.3	+26.1	--- ^a^
3′	+14.9	+15.2	+4.31
1	+12.3	+13.7	--- ^a^
10′	+10.3	+9.0	+4.10
10	+7.3	+6.3	+4.36
5	+2.6	+1.9	--- ^b^
1′	+2.5	+1.2	+3.41
6	+0.7	−0.7	--- ^b^
9	+0.7	−2.7	+3.47
9′	+0.7	−2.9	+3.39
8	−2.4	−4.8	+3.09
8′	−5.3	−7.6	+3.34
7	−10.3	−12.7	+3.09
7′	−14.3	−15.9	+3.27
2	−31.4	−34.6	+0.33
2′	−31.9	−34.8	+0.33
4	−38.6	−42.1	+0.68
4′	−38.8	−41.2	+0.68
9′,10′			−0.10
9,10			−0.19
5,7′			−0.43
6,7			−0.90
8′,9′			−2.44
8,9			−2.80

^a^ Chlorine substituent, ^b^ Site of conjunction.

**Table 6 molecules-28-04505-t006:** Measured boron-11 NMR data for 7,3′-Cl_2_-B_18_H_20_ (**6**) at 291 K in CDCl_3_ solution together with calculated values for comparison. (^11^B and ^11^B-{^1^H} NMR spectra are shown in Appendix A).

Assign.	δ(^11^B)/ppm	
	Meas.	Calc.
3′ ^a^	+23.4	+23.6
3	+14.0	+11.8
10′	+10.0	+10.3
10	+6.8	+7.0
5 ^b^	+3.7	+3.1
7 ^a^	+3.1	+3.6
9	+2.9	+3.6
1′	+1.4	+1.5
9′	+1.4	+1.8
1	−0.2	−1.5
6 ^b^	−1.0	−1.7
8	−1.5	−1.7
8′	−4.4	−4.9
7′	−14.3	−13.2
2′	−31.5	−30.5
2	−32.8	−32.0
4′	−38.7	−37.0
4	−40.5	−39.1

^a^ Chlorine substituent, ^b^ Site of conjunction.

**Table 7 molecules-28-04505-t007:** Measured proton and boron-11 NMR data for 3-Cl-B_18_H_21_ (**7**) at 291 K in CDCl_3_ solution together with calculated values for comparison. (^11^B and ^11^B-{^1^H} NMR spectra are shown in Appendix A).

Assign.	δ(^11^B)/ppm		δ(^1^H)/ppm
	Meas.	Calc.	Meas.
3	+23.2	+26.1	--- ^a^
3′	+13.5	+13.2	+3.89
10	+9.8	+9.0	+4.09
10′	+8.3	+7.6	+4.08
6	+4.5	+4.0	--- ^b^
1	+3.0	+0.2	+3.42
5	+1.5	+0.1	--- ^b^
9′	+0.7	−1.8	+3.36
1′	+0.6	−1.8	+3.37
9	−0.4	−3.0	+3.10
8′	−4.4	−6.5	+3.06
8	−5.6	−8.0	+3.29
7′	−11.1	−13.2	+0.09
7	−13.3	−15.2	
2	−31.9	−34.6	
2′	−33.6	−36.3	
4	−39.3	−42.1	
4′	−40.2	−43.5	

^a^ Chlorine substituent, ^b^ Site of conjunction.

**Table 8 molecules-28-04505-t008:** Calculated zero-point corrected energies for the seven most stable Cl_3_-B_18_H_19_ isomers relative to 1,3,3′-Cl_3_-B_18_H_19_ as the lowest energy isomer. Associated colour code indicates the extent to which zero-point corrected energies for individual isomers increase—from green (lowest energy) to red (highest energy).

Isomer	kcal/mol
1,3,3′-Cl_3_-B_18_H_19_	0.0
1,3,1′-Cl_3_-B_18_H_19_	0.3
3,4,3′-Cl_3_-B_18_H_19_	1.1
1,4,3′-Cl_3_-B_18_H_19_	1.4
1,3′,4′-Cl_3_-B_18_H_19_	1.5
1,4,1′-Cl_3_-B_18_H_19_	1.6
3,4,4′-Cl_3_-B_18_H_19_	2.5

**Table 9 molecules-28-04505-t009:** Measured boron-11 NMR data for 3,4,3′-Cl_3_-B_18_H_19_ (**9**) at 291 K in CDCl_3_ solution together with calculated values for comparison. (^11^B and ^11^B-{^1^H} NMR spectra are shown in Appendix A).

Assign.	Δ(^11^B)/ppm	
	Meas.	Calc.
3′ ^a^	+22.1	+19.4
3 ^a^	+21.3	+18.6
10′	+8.5	+5.9
10	+8.5	+5.5
1	+0.6	−3.1
9′	+0.1	−3.6
5 ^b^	−0.1	−4.1
9	−0.1	−4.3
1′	−0.1	−4.6
6 ^b^	−0.1	−4.8
8	−5.9	−9.4
8′	−5.9	−9.5.7
7′	−12.8	−15.0
7	−13.5	−15.6
4 ^a^	−25.4	−25.4
2	−33.8	−35.3
2′	−34.2	−35.7
4	−40.1	−41.5

^a^ Chlorine substituent, ^b^ Site of conjunction.

**Table 10 molecules-28-04505-t010:** Photophysical properties of the chlorinated derivatives of *anti*-B_18_H_22_ in air-saturated hexane at room temperature ^a^.

	*λ_abs_*/nm	*λ_L_*/nm	*τ_L_*/ns	*Φ_L_* (λ_exc_/nm)
*anti*-B_18_H_22_ (1) ^b^	335	406	11.2	0.97 (335 nm)
7-Cl-B_18_H_21_ (11) ^c^	344	418	8.3	0.80 (340 nm)
3-Cl-B_18_H_21_ (7)	327	407	6.1	0.52 (320 nm)
4,4′-Cl_2_-B_18_H_20_ (4)	344	435	1.5	0.17 (340 nm)
3,3′-Cl_2_-B_18_H_20_ (2)	324	410	1.2	0.07 (320 nm)

^a^ *λ_abs_*—absorption maximum; *λ_L_—*emission maximum (λ_exc_ = 320 nm); *τ_L_*—fluorescence lifetime in air-saturated hexane (λ_exc_ = 340 nm, λ_em_ = 450 nm,); *Φ_L_*—fluorescence quantum yield in air-saturated hexane (experimental error of *Φ_L_* is ±0.01). ^b^ data taken from reference [4]. ^c^ data taken from reference [13].

**Table 11 molecules-28-04505-t011:** Crystal and refinement data for compounds **2**–**8**.

	2	3	4	5	6	7/8
Chemical formula	B_18_H_20_Cl_2_	B_18_H_20_Cl_2_	B_18_H_20_Cl_2_	B_18_H_20_Cl_2_	B_18_H_20_Cl_2_	B_18_H_21_Cl
*M* _r_	285.6	285.6	285.6	285.6	285.6	251.2
Crystal system	monoclinic	orthorhombic	triclinic	monoclinic	monoclinic	monoclinic
Space group	*P*2_1_/*c*	*Aba*2	*P* $\bar{1}$	*P*2_1_	*C*2/*c*	*P*2_1_/*c*
*a* (Å)	10.98586(12)	18.9603(5)	11.6342(9)	7.2776(2)	14.1953(3)	7.2553(3)
*b* (Å)	12.76094(15)	14.4383(3)	11.6732(9)	10.6035(2)	11.8073(2)	10.8304(4)
*c* (Å)	11.21906(13)	12.0489(2)	11.6840(8)	10.4842(3)	20.3684(4)	10.0028(3)
α (°)			89.695(6)			
β (°)	94.9949(10)		86.618(6)	100.525(4)	112.580(2)	106.432(3)
γ (Å)			84.082(6)			
*V* (Å^3^)	1566.83(3)	3298.44(12)	1575.6(2)	795.43(4)	3152.21(11)	753.89(5)
*Z*	4	8	4	2	8	2
Crystal size (mm^3^)	0.21 × 0.17 × 0.12	0.10 × 0.06 × 0.01	0.09 × 0.08 × 0.02	0.14 × 0.10 × 0.06	0.12 × 0.07 × 0.05	0.09 × 0.06 × 0.02
*T* (K)	100	150	100	100	100	150
Radiation, λ (Å)	Cu*K*α, 1.54184	Cu*K*α, 1.54184	Cu*K*α, 1.54184	Mo*K*α, 0.71073	Cu*K*α, 1.54184	Cu*K*α, 1.54184
Reflections measured	20,387	15,233	14,596	11,545	10,723	7108
Unique reflections	3301	2229	6177	3144	3315	1504
*R* _int_	0.0432	0.0347	0.1203	0.0301	0.0248	0.0272
Parameters, restraints	261, 0	260, 2	482, 772	282, 79	261, 0	139, 1
*R* (*F*, *F*^2^ > 2σ)	0.0276	0.0362	0.1626	0.0721	0.0340	0.0597
*R*_w_ (*F*^2^, all data)	0.0719	0.1027	0.4501	0.2246	0.0977	0.1568
Goodness of fit (*F*^2^)	1.121	1.046	1.065	1.095	1.069	1.114
Flack parameter				0.02(5)		
Max, min Δρ (e Å^−3^)	0.31, −0.24	0.42, −0.16	1.86, −0.69	0.33, −0.40	0.61, −0.26	0.21, −0.22
CCDC deposition number	2256510	2256511	2256512	2256513	2256514	2256515

## Data Availability

DFT calculations for the chlorinated boranes described here were performed using the Gaussian 09 package [24] at the B3LYP/6-311++G(d,p) level for all atoms. Frequency analyses were carried out to confirm the validity of minima. GIAO NMR nuclear shielding predictions were performed at the appropriate level on the final optimised geometries and the calculated boron nuclear shielding values were related to chemical shifts δ(^11^B) by comparison with the computed value for B_2_H_6_ which was taken to be δ(^11^B) +16.6 ppm relative to the F_3_B.OEt_2_ at 0.0 ppm. Zero-point corrected energy calculations were performed using the Gaussian 16W package source at the B3LYP/cc-pVDZ level for all atoms.

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
