# Peer review of "A Window into the Workings of anti-B18H22 Luminescence—Blue-Fluorescent Isomeric Pair 3,3′-Cl2-B18H20 and 3,4′-Cl2-B18H20 (and Others)†"

_molecules, 2023, doi:10.3390/molecules28114505_

Round 1

Reviewer 1 Report

The work ''A window into the working of anti-B18H22 luminescence. Blue-fluorescent isomeric pair 3,3'-Cl2-B18H20 and 3,4'-Cl2-B18H20 (and others)'' by Marcel Ehn et.al describes the chemical modification of boron hydride cluster B18H22 by AlCl3 and characterisation of 6 new compounds with single crystal X-ray crystallogpraphy. The emission spectra among with the fluorescent decay spectra of the compounds 2, 4 and 7 gave information about the influence of the Cl ligands in the B18 cluster, leading to different lifetimes depending on the Cl position in B cluste sites. The work is well written and the results are clearly presented. I believe that this work should be accepted for publication in MDPI-Molecules as is. A very minor suggestion could be a more extended Introduction Section as for a reader without the knowledge of the field (like me) to be more familiarised with the work.

The ms describes new B18H22 derivatives derived from the reaction between anti-B18H22 and AlCl3 in CCl4. These compounds are added to a family of substituted B18H22 clusters, with Cl occupying different positions in the 18-vertex B cluster. These compounds have been characterized using single crystal X-ray crystallography, and in the ms are also presented the emission and lifetime decay spectra. Also, the authors claim that these compounds may possess interesting laser performance.

 This manuscript adds new compounds to the family of substituted B18H22 clusters, so I believe that is relevant to the field of B compounds.

The conclusions can also be improved and extended regarding the application of the new compounds.

The quality of the English is good.

Author Response

The reviewer's comments are complimentary of the work and suggests publication as is, but recommends extension of Introduction and Conclusion sections.

In response, we have widened the Introduction to include a new passage that explains to the reader the context of the boron hydrides and 'macropolyhedral' clusters, including two new references (Refs 1 and 2) that are perfect for non-specialist readers to cover the fundamentals if interested:

"With the obvious exception of carbon, it is boron that amongst all elements boasts the greatest number and diversity of hydride compounds.1 These species produce polyhedral cluster assemblies with an ostensible 12-vertex icosahedral size limit. This size constraint, however, can be surpassed by the intimate fusion of two or more clusters to form ‘macropolyhedral’ species in which constituent subclusters conjoin via shared polyhedral edges or faces.2"

  1. Beall, H.; Gaines, D. F. Boron Hydrides, Encyclopedia of Physical Science and Technology (3rd Edition, Academic Press, 2003, 301-316.
  2. Shea, S. L.; Bould, J.; Londesborough, M. G. S.; Perera, S. D.; Franken, A.; Ormsby, D. L.; Jelínek, T.; Štíbr, B.; Holub, J.; Kilner, C. A.; Thornton-Pett, M.; Kennedy, J. D. Polyhedral boron-containing cluster chemistry: Aspects of architecture beyond the icosahedron. Pure and Applied Chemistry, 2003, vol. 75, no. 9, pp. 1239-1248.

Regarding the Conclusion, we do not believe that we could expand on our current text without further work.  We end on the reasonable speculation that if we are able to combine the shortening effect on the lifetime of excited-states that chlorination enacts on B18H22, with the superior absorption and emission properties that alkylation induces, then a "sweet spot" may be found for laser emission.  To test this hypothesis, which is our intention, requires a lot of work (synthetic, structural and analytical) and would be the subject of a further publication.  Thus, we ask the reviewer here for kind understanding that our conclusions cannot stretch further, at this point, than what is already written.

Reviewer 2 Report

This work deals with a rather new class of substituted anti-B18H22 compounds. The interest in such compounds is dictated primarily by the fact that they exhibit good luminescent properties and can be used in laser and optical devices.

The work is of high quality and involves a painstaking technique of isolating isomeric products. It is also worth noting that almost all the products are characterized by XRD.

There are a few comments on the manuscript.

 1. We ask the authors to revise the tables with NMR spectroscopy data. Perhaps present them as figures, as in the current format these data are difficult to grasp.

2. Add to the discussion part a more detailed description of the crystal structures. Provide a comparison of bond lengths and geometry depending on the number and position of substituent. Perhaps a summary table should be given.

3. Redo Figure 3.

 After making the necessary corrections, the article can be published in Molecules.

Author Response

We are glad that the reviewer has an overall complimentary appraisal of our manuscript, and we have addressed his/her direct points in the following manner:

  1. We ask the authors to revise the tables with NMR spectroscopy data. Perhaps present them as figures, as in the current format these data are difficult to grasp.

The NMR data in Table form are very useful for synthetic chemists repeating the work.  So, we believe that this format should remain.  All recorded NMR spectra are in the SI for easier visualisation.   However, we have generated and included a new Figure 2 in the manuscript that displays, in a stacked formation for easy comparison, the 11B-{1H} for all the new chlorinated derivatives of B18H22 described.  We hope that this helps the Reviewer and all readers in grasping the data more easily.

2. Add to the discussion part a more detailed description of the crystal structures. Provide a comparison of bond lengths and geometry depending on the number and position of substituent. Perhaps a summary table should be given.

We have done a thorough detailed analysis of the B-B bond lengths in the ordered structures of this series, in response to the reviewer's suggestion.  The conclusion is that changing the site of chlorination has essentially no impact on the cluster geometry; observed differences from the parent anti-B18H22 (for which there's a very precise structure in the CSD, taken from a 2006 paper in Chem. Eur. J. that also describes a benzene solvate) are relatively small compared with the overall range of B-B bond lengths and appear to be more or less random; there's as much variation among crystallographically independent but chemically identical molecules as these differences.

Into the manuscript we have included the following new text:

"A detailed comparison of the molecular geometry of these various chlorinated clusters, together with the parent anti-B18H22 (1),18 shows that the B–B (and B–Cl) bond lengths are rather insensitive to the substituent positions.  Because structural disorder has an averaging effect on parts of the molecules in which disorder is not resolved and twinning adversely affects precision, such an analysis can be carried out reliably only on ordered structures, which here means one form of compound 2 (not the forms in the Supporting Information), and compounds 3, 4, and 6; two of these (2 and 4) have more than one crystallographically independent molecule, all of which can be included in the comparison.  A full list of bond lengths, and their differences from the corresponding bonds in compound 1, are given in Table S38.  These differences between the substituted and parent clusters range from −0.045 Å to +0.034 Å, while the total range of all B–B bond lengths is 0.294 Å (from 1.706 to 2.002 Å, with individual crystallographic standard uncertainties between about 0.002 and 0.02 Å); they follow no clear pattern relative to the positions of substitution.  We therefore conclude that the cluster bonding is essentially unaffected by chlorination."

3. Redo Figure 3.

This we have done.